# Development and Validation of a Radiomics Nomogram for Liver Metastases Originating from Gastric and Colorectal Cancer

**DOI:** 10.3390/diagnostics13182937

**Published:** 2023-09-13

**Authors:** Yuying Li, Jingjing Li, Mingzhu Meng, Shaofeng Duan, Haifeng Shi, Junjie Hang

**Affiliations:** 1Department of Radiology, Changzhou Second People’s Hospital Affiliated with Nanjing Medical University, Changzhou 213000, China; 13824554947@163.com (Y.L.); l15327963866@163.com (J.L.); mengmz0202@njmu.edu.cn (M.M.); 2Graduate College, Dalian Medical University, Dalian 116000, China; 3GE Healthcare, Precision Health Institution, Shanghai 201100, China; 18910063803@163.com; 4Department of Oncology, National Cancer Center/National Clinical Research Center for Cancer/Cancer Hospital & Shenzhen Hospital, Chinese Academy of Medical Sciences and Peking Union Medical College, Shenzhen 518116, China; 5Department of Oncology, Changzhou Second People’s Hospital Affiliated with Nanjing Medical University, Changzhou 213000, China

**Keywords:** liver, neoplasm metastasis, radiomics, computed tomography, stomach neoplasms, colorectal neoplasms

## Abstract

The origin of metastatic liver tumours (arising from gastric or colorectal sources) is closely linked to treatment choices and survival prospects. However, in some instances, the primary lesion remains elusive even after an exhaustive diagnostic investigation. Consequently, we have devised and validated a radiomics nomogram for ascertaining the primary origin of liver metastases stemming from gastric cancer (GCLMs) and colorectal cancer (CCLMs). This retrospective study encompassed patients diagnosed with either GCLMs or CCLMs, comprising a total of 277 GCLM cases and 278 CCLM cases. Radiomic characteristics were derived from venous phase computed tomography (CT) scans, and a radiomics signature (RS) was computed. Multivariable regression analysis demonstrated that gender (OR = 3.457; 95% CI: 2.102–5.684; *p* < 0.001), haemoglobin levels (OR = 0.976; 95% CI: 0.967–0.986; *p* < 0.001), carcinoembryonic antigen (CEA) levels (OR = 0.500; 95% CI: 0.307–0.814; *p* = 0.005), and RS (OR = 2.147; 95% CI: 1.127–4.091; *p* = 0.020) exhibited independent associations with GCLMs as compared to CCLMs. The nomogram, combining RS with clinical variables, demonstrated strong discriminatory power in both the training (AUC = 0.71) and validation (AUC = 0.78) cohorts. The calibration curve, decision curve analysis, and clinical impact curves revealed the clinical utility of this nomogram and substantiated its enhanced diagnostic performance.

## 1. Introduction

The term “Cancer of Unknown Primary” (CUP) denotes a diverse cohort of metastatic cancers wherein the tissue of origin eludes identification following a conventional diagnostic assessment [1,2,3,4]. CUPs constitute approximately 2–5% of all human malignancies [3,4]. The prevailing histological pattern observed in most CUP cases manifests as moderately differentiated adenocarcinomas, although instances of undifferentiated or poorly differentiated adenocarcinomas, squamous cell carcinomas, neuroendocrine carcinomas, or undifferentiated neoplasms also exist [2,3,4]. Smoking may be regarded as a potential risk factor for malignancies arising from unknown primary sites [5]. Moreover, the overall prognosis for CUP patients typically presents a grim outlook, with a 12-month survival rate of merely 24% [6]. Nevertheless, specific subtypes, such as lymphomas, extragonadal germ cell tumours, and neuroendocrine tumours, hold the potential for curative interventions [2,3]. It is worth noting that CUP can occasionally manifest as prostate cancer, for which well-established treatment protocols are available [4]. Generally, patients who eventually have an identifiable primary tumour or those with tumours that demonstrate sensitivity to chemotherapy or hormone therapy tend to exhibit a more favourable prognosis [4].

The two most prevalent types of hollow organ tumours are gastric and colorectal cancers, ranking as the third and sixth most commonly diagnosed malignancies, respectively. Notably, they both persist as leading causes of cancer-related fatalities worldwide [7,8]. The liver emerges as one of the most frequently affected sites for metastatic dissemination in cases of hollow organ tumours, owing to its dual blood supply from the hepatic artery and portal vein [7]. The presence of liver metastases arising from hollow organ tumours, such as those originating in the stomach and colorectum, can yield a more favourable prognosis, primarily due to the array of available treatment modalities. For instance, hepatectomy is recommended for patients with liver metastases from gastric cancer (GCLMs) presenting with three or fewer liver metastases or unilobar metastases [8]. In contrast, hepatectomy is generally indicated for liver metastases from colorectal cancer (CCLMs), regardless of their number or distribution [9]. Furthermore, patients with GCLMs exhibit significantly shorter times to surgical failure (median, 15.2 months vs. 39.7 months, *p* = 0.006) and overall survival (median, 20.1 months vs. 66.2 months, *p* < 0.001) when compared to those with CCLMs. Previous investigations have established that GCLMs tend to represent a more systemic disease than CCLMs, justifying the rationale for administering chemotherapy to GCLM patients [10]. Nevertheless, there are instances where the primary lesion remains elusive despite extensive diagnostic scrutiny, particularly in the context of hollow organ tumours [11,12,13]. Gastrointestinal wall thickness, assessable via computed tomography (CT) scans, can increase due to benign factors, such as inflammation, ulcers, polyps, tuberculosis, Crohn’s disease, diverticula, or Menetrier’s disease [14]. In addition, 18F-FDG positron emission tomography (PET)/CT exhibits limited diagnostic utility in detecting primary hollow organ tumours [15,16]. Determining the primary site is an invasive, time-consuming, and costly endeavour. Consequently, the development of a non-invasive, straightforward, and effective model for discerning the primary site of hollow organ tumours, specifically gastric and colorectal cancers, is of paramount importance.

Radiomics entails the precise conversion of image-derived information into accessible data for the automated extraction of quantitative features [17,18]. Lang et al. [17] achieved successful differentiation between metastatic spinal lesions originating from primary lung cancer and those from other cancer types using radiomics. Several studies [18,19,20] have reported the capacity of radiomics to distinguish brain metastases originating from breast, lung, and various other cancer categories. However, liver metastases have received relatively scant attention in this context. Ben-Cohen et al. [21] conducted a study involving 71 patients with liver metastases, employing CT data as the input source to classify primary sites. Similarly, Qin et al. [22] retrospectively examined 254 patients, constructing and validating radiomics models based on B-mode ultrasound features for identifying the origins of hepatic metastatic lesions. Both investigations demonstrated that texture analysis could facilitate the differentiation of liver metastases arising from distinct primary sites. Nonetheless, further research is warranted, given the limited dataset and the propensity for overfitting. Additionally, gastric and colorectal cancers, although originating from the same bodily system, albeit different organs, have been the subject of relatively few comparative studies in this context.

Hence, the objective of this study is to develop and validate a radiomics nomogram for the precise determination of the primary site of liver metastases originating from these two cancer types.

## 2. Study Design and Patients

This study retrospectively enrolled patients diagnosed with either GCLMs or CCLMs at Changzhou Second People’s Hospital, affiliated with Nanjing Medical University, during the period spanning from April 2011 to September 2021. Approval for this retrospective study was obtained from the Institutional Review Board of Changzhou Second People’s Hospital, affiliated with Nanjing Medical University. Given the retrospective nature of the study, the requirement for informed consent was waived.

The inclusion criteria encompassed the following conditions: (1) newly diagnosed cases of gastric or colorectal cancer with confirmed liver metastases through pathological examination; (2) the absence of concurrent cancers in other organ sites; (3) liver lesions confirmed as metastases either via biopsy or, for patients lacking histopathological data, by displaying typical metastatic imaging characteristics and morphological changes (involving at least a 30% increase or reduction in maximum diameter) during the follow-up period; (4) undergoing dynamic abdominal CT scans as part of their treatment regimen; (5) no prior history of radiotherapy, chemotherapy, or other treatments; and (6) comprehensive baseline clinicopathological and epidemiological data, including age, sex, hypertension, diabetes, and other relevant factors. On the other hand, the exclusion criteria comprised the following: (1) cases where the primary malignant lesion remained unidentified; (2) the presence of concurrent cancers in other organ sites; (3) a lack of morphological changes in liver metastases during follow-up after treatment; (4) instances where CT images were lost or the image quality was deemed suboptimal; or (5) incomplete clinical data.

### 2.1. Data Collection and Definition

Demographic and clinicopathological data were retrieved from the electronic medical records of patients, encompassing information such as age, gender, the presence of hypertension, diabetes, the existence of liver metastases, primary site tumour location, haemoglobin (HGB) levels, alanine aminotransferase (ALT) levels, carcinoembryonic antigen (CEA) levels, and cancer antigen (CA)19-9 levels. The primary site tumours under consideration included gastric cancer and colorectal cancer.

### 2.2. CT Image Acquisition and Image Processing

Contrast-enhanced CT examinations were conducted using a 128-row dual-source CT scanner (SOMATOM Definition Flash, Siemens, Germany), employing a voltage of 120 kV and tube current modulation. All patients were instructed to fast for a minimum of 8 h before receiving intravenous contrast (Iohexol, 1.5 mL per kilogram of body weight, administered at a rate of 3 mL/s). Following the administration of the contrast agent, patients underwent double-helical scanning during both the arterial and portal venous phases.

Regions of interest (ROIs) were delineated on each slice displaying liver metastases during the portal venous phase using the open-source imaging platform LabelMe software (version 3.11.2, https://github.com/wkentaro/labelme, accessed on 22 March 2019). Subsequently, all ROIs were imported into freeware, specifically the Local Image Features Extraction (LIFEx) software (version 5.10, https://www.lifexsoft.org, accessed on 19 March 2018), for texture analysis. To compute first-order features for the segmented tumours [23], a histogram was generated. As detailed in prior studies [23,24], the number of grey levels used for resampling the ROI content was set at 64.0 to calculate second-order and higher texture features. Spatial resampling was configured at 2.0 mm (X direction), 2.0 mm (Y direction), and 1.0 mm (Z direction) in Cartesian coordinates. Texture features were evaluated via four texture matrices, namely the grey-level co-occurrence matrix (GLCM), the grey-level run length matrix (GLRLM), the neighbourhood grey-level difference matrix (NGLDM), and the grey-level zone length matrix (GLZLM). In total, 35 features were extracted from the texture analysis. The features from the largest ROI of each patient were employed for subsequent analysis. The radiomics signature (RS) was computed based on feature variations, with RS = −1.191 × Skewness + 1.223 × Homogeneity + −1.2 × Dissimilarity. In this formula, each variable (*p* < 0.10) was weighted using its β-coefficient derived from the univariable analyses.

### 2.3. Nomogram Development and Validation

All patients underwent random allocation into training and validation cohorts, adhering to a 7:3 ratio. Initially, only the training cohort was utilised for the selection of predictive features. Variables exhibiting a significance level of *p* < 0.10 in the univariable analysis were subsequently incorporated into the multivariable logistic regression analysis. The reference group for this analysis was the CCLM group. Subsequently, we developed a nomogram employing the R package “rms” within the R software (version 3.6.1, Institute for Statistics and Mathematics, Vienna, Austria). This nomogram aimed to distinguish the primary tumour origin, discerning between colorectal and gastric cancers based on the outcomes of the multivariable logistic regression analysis. The predictive performance of the nomogram was quantified using the area under the curve (AUC) from the receiver operating characteristic (ROC) curve analysis, conducted in both the training and validation cohorts, utilising the R package “ROCR”. Internal validation of the nomogram was executed via calibration curve analysis, employing bootstrapping with 1000 resamples to assess the consistency between actual outcomes and predicted probabilities. Lastly, the clinical utility of the nomogram was evaluated through decision curve analysis (DCA) and clinical impact curve (CIC) assessment, considering a population size of 1000.

### 2.4. Statistical Analysis

Statistical analysis was performed using R (version 3.6.1, Institute for Statistics and Mathematics, Vienna, Austria) and SPSS (version 22.0, IBM Corp., Armonk, NY, USA). Student’s *t*-test and the chi-square test were applied to assess differences in continuous and categorical variables, respectively, between groups. The correlations among texture parameters were evaluated using Pearson’s correlation coefficient, facilitated by the R package “psych”. To investigate disparities in texture features between patients with GCLMs and those with CCLMs, a two-sided Student’s *t*-test was employed. Statistical significance was defined as two-sided *p*-values < 0.05.

## 3. Results

### 3.1. Characteristics of the Patients

A total of 555 patients were included in the study, comprising 277 with GCLMs and 278 with CCLMs. The training cohort comprised 400 patients, while the validation cohort included 155 patients. No statistically significant differences in clinical characteristics were observed between the training and validation cohorts (all *p* > 0.05, as indicated in Table 1).

### 3.2. Clinical Characteristics between Patients with GCLMs and CCLMs

In the training cohort, when comparing patients with CCLMs to those with GCLMs, several noteworthy differences emerged. Notably, the CCLM group exhibited a lower proportion of males (*p* < 0.001), higher levels of haemoglobin (HGB) (*p* < 0.001), elevated carcinoembryonic antigen (CEA) levels (*p* = 0.039), and a reduced radiomics signature (RS) (*p* = 0.010). Conversely, no statistically significant disparities were identified in other factors, including age, hypertension, diabetes mellitus, the number of liver metastases, alanine aminotransferase (ALT), or cancer antigen 19-9 (CA19-9) (all *p* > 0.05). In the validation cohort, a similar pattern was observed when comparing patients with CCLMs to those with GCLMs. Specifically, the CCLM group exhibited a lower proportion of males (*p* < 0.001), higher HGB levels (*p* = 0.053), lower ALT levels (*p* = 0.021), elevated CEA levels (*p* = 0.007), and a higher RS (*p* = 0.039) (Table 2).

### 3.3. Correlation between Texture Parameters of Liver Metastases in the Training Set

Within the training cohort, no statistically significant distinctions were evident when comparing the GLRLM, NGLDM, or GLZLM parameters between the two groups, as determined by Student’s *t*-test (all *p* > 0.10) (Table 3). In the context of histogram analysis, patients with GCLMs displayed a trend towards higher skewness values in comparison to those with CCLMs (*p* = 0.097). A similar pattern emerged in the GLCM analysis, wherein patients with GCLMs exhibited greater levels of homogeneity (*p* = 0.078) and dissimilarity (*p* = 0.096) than their CCLM counterparts. To investigate the relationships between texture parameters in the training cohort, Pearson’s correlation coefficient was employed (Figure 1).

### 3.4. Development and Validation of the Nomogram

The multivariable regression analysis demonstrated that sex (OR = 3.457; 95% CI: 2.102–5.684; *p* < 0.001), haemoglobin (HGB) levels (OR = 0.976; 95% CI: 0.967–0.986; *p* < 0.001), carcinoembryonic antigen (CEA) levels (OR = 0.500; 95% CI: 0.307–0.814; *p* = 0.005), and the radiomics signature (RS) (OR = 2.147; 95% CI: 1.127–4.091; *p* = 0.020) stood as independent factors associated with GCLMs relative to CCLMs (Table 4). To offer a visual representation of the predictive model, a nomogram was constructed, incorporating these independent predictors (Figure 2). The area under the receiver operating characteristic (ROC) curve for the training and validation cohorts was calculated as 0.71 and 0.78, respectively (Figure 3). In the training cohort, the nomogram exhibited a specificity and sensitivity of 77.66% and 55.56%, respectively, with a positive predictive value of 71.43% and a negative predictive value of 63.48%. In the validation cohort, the nomogram demonstrated a specificity of 66.7%, sensitivity of 79.2%, positive predictive value of 70.12%, and negative predictive value of 76.47%. Calibration curve analysis of the training cohort indicated a relatively strong alignment between the predicted and actual probabilities of the nomogram (Figure 4). Decision curve analysis (DCA) in the validation cohort revealed that utilising the nomogram to predict the likelihood of GCLMs would yield greater clinical benefits if the threshold probability ranged from 18% to 82% (Figure 5A). Clinical impact curve (CIC) analysis demonstrated that as the threshold probability increased, the number of high-risk cases also rose, underscoring the valuable clinical utility of the nomogram in patients with GCLMs (Figure 5B).

## 4. Discussion

In this study, we developed and validated a nomogram that incorporates sex, haemoglobin (HGB) levels, carcinoembryonic antigen (CEA) levels, and the radiomics signature (RS) to predict the origin of liver metastases, specifically distinguishing between gastric cancer and colorectal cancer. The nomogram exhibited robust discrimination capabilities in both the training and validation cohorts. These features hold significant clinical relevance as they can enhance the ability of healthcare providers to predict the primary tumour site and make informed decisions regarding subsequent treatment strategies.

In practical clinical scenarios, the assessment of the primary tumour site should prioritise convenience and efficiency while avoiding unnecessary invasive procedures that do not substantially benefit patients. Radiomics, as a noninvasive technology, shows immense promise in delivering results comparable to invasive methods. Our radiomics model was successfully developed and demonstrated remarkable proficiency in distinguishing between gastric cancer liver metastases (GCLMs) and colorectal cancer liver metastases (CCLMs). Notably, our study observed a higher prevalence of GCLMs in male patients compared to CCLMs, aligning with previous findings by Chen et al. [25]. Furthermore, patients with GCLMs displayed a higher incidence of anaemia, which may be attributed to reduced gastric acid and intrinsic factor production, leading to the impaired absorption of micronutrients like iron and vitamin B12, ultimately resulting in anaemia [26,27]. Additionally, Ning et al. [28] reported elevated average CEA levels in patients with colorectal cancer in comparison to those with gastric cancer (mean CEA: 50.36 vs. 23.78 U/mL). This observation suggests that CEA could potentially serve as an independent predictor for both gastric and colorectal cancer, supporting our study’s findings.

Texture analysis, relying on imaging techniques, facilitates the evaluation of tumour heterogeneity by examining the distribution and interrelationships of grey levels at the pixel or voxel level within an image [29]. Previous investigations have delved into the association between radiomics features and clinicopathological data. Weber et al. [30] observed a positive correlation between dissimilarity and Ki-67, while homogeneity exhibited an inverse relationship with Ki-67. Furthermore, they demonstrated the potential to differentiate the pathological grading of liver metastases by utilising measures of homogeneity and dissimilarity. Martini et al. [31] identified that non-pancreatic neuroendocrine tumours displayed higher skewness in liver metastases compared to pancreatic neuroendocrine tumours. Additionally, higher baseline entropy and lower homogeneity in liver metastases were linked to improved survival rates and enhanced responses to chemotherapy. Homogeneity also emerged as a robust predictor of tumour regression grade [32].

In our study, we selected Histogram_Skewness, GLCM_Homogeneity, and GLCM_Dissimilarity of liver metastases as valuable predictors to construct the radiomics signature (RS). Skewness, measuring the asymmetry of the histogram, serves as a first-order feature indicating intra-tumour heterogeneity in terms of intensity [33]. Previous research has established a connection between lower skewness and reduced angiogenesis, suggesting a higher likelihood of tumour hypoxia and necrosis [34]. Homogeneity, reflecting the uniformity of the co-occurrence matrix, and dissimilarity, a measure of differences between elements in the matrix, are second-order texture features based on the grey-level co-occurrence matrix (GLCM). These features capture the heterogeneity of the spatial distribution of voxel intensities. Tumours demonstrating greater homogeneity (higher homogeneity and lower dissimilarity) may signify more compact necrotic areas [35]. Hypoxia and necrosis can elevate interstitial hydrostatic pressure in the tumour microenvironment, increasing the risk of tumour invasiveness and metastasis [36]. Our study revealed that GLCMs exhibited lower skewness, dissimilarity, and higher homogeneity than CCLMs, suggesting heightened heterogeneity in the former. These findings align with prior research [32,37,38,39]. Oguro et al. [10] reported that GLCMs exhibited a more systemic and aggressive oncological behaviour than CCLMs, further substantiating the utility of the aforementioned texture features in distinguishing the primary site (gastric and colorectal cancer) of liver metastases.

We devised a radiomics nomogram capable of predicting the specific origin sites, namely gastric and colorectal cancer. This nomogram effectively gauges the likelihood of the primary tumour originating from either gastric or colorectal cancer. Furthermore, we evaluated the clinical utility of these nomograms using decision curve analysis (DCA) and clinical impact curve (CIC), revealing their efficacy across a wide range of threshold probabilities. This tool can assist clinicians in identifying cases of gastric cancer liver metastases (GCLMs) as potential site-specific malignancies based on probability estimates. Subsequently, tailored site-specific therapies can be individually tailored by referencing established treatment protocols for the known primary tumour site, potentially benefiting these patients [40].

However, it is important to acknowledge the limitations of this study. Firstly, we performed texture analysis solely on the largest cross-sectional area of the tumour, rather than conducting a comprehensive analysis of the entire tumour volume. It is worth noting that a prior study demonstrated comparable results when focusing on the largest cross-sectional area [41]. Secondly, the predictive model developed in this study is specific to gastric and colorectal cancer and does not encompass other types of primary tumours. Nonetheless, distinguishing the primary site of tumours affecting hollow organs poses a prevalent and significant challenge for radiologists and clinicians in their routine medical practice. Thirdly, it is crucial to recognise that this investigation is retrospective in nature and was conducted within a single medical centre. External validation is imperative to further confirm the clinical effectiveness of the nomogram established in this research. Fourthly, while the area under the curve (AUC) results were favourable, they were not perfect, indicating room for potential model enhancement. Lastly, it is important to note again that this study was retrospective, limiting the analysis to available data in the patient charts, precluding an assessment of treatments and prognoses.

## 5. Conclusions

In this study, we developed a CT-based radiomics nomogram to discern the primary site of liver metastases originating from gastric cancer and colorectal cancer. Our preliminary findings indicate that the nomogram, comprising sex, HGB, CEA, and RS, exhibits excellent discriminatory ability and holds significant clinical value for distinguishing between GCLMs and CCLMs. This nomogram can serve as a valuable reference tool for accurate diagnoses by radiologists and clinicians, streamlining the selection of appropriate and timely treatments. Moreover, the results may provide valuable insights for future radiomics investigations.

## Figures and Tables

**Figure 1 diagnostics-13-02937-f001:**
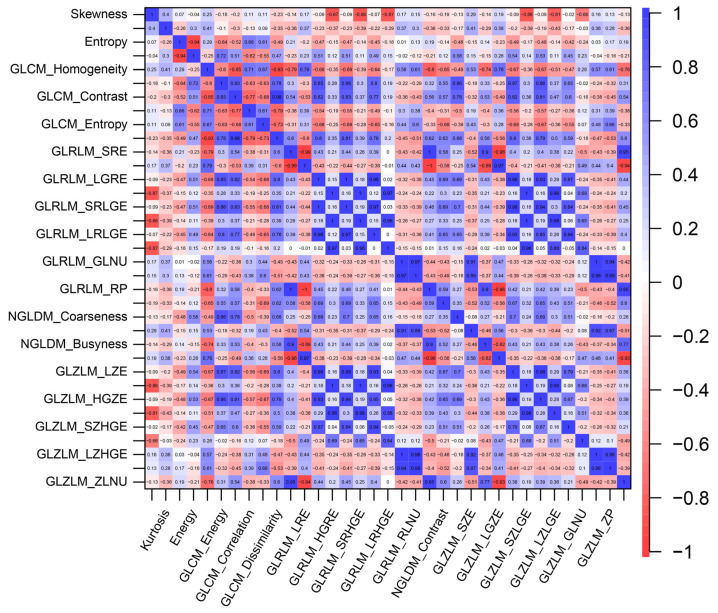
The correlations between texture parameters within liver metastases.

**Figure 2 diagnostics-13-02937-f002:**
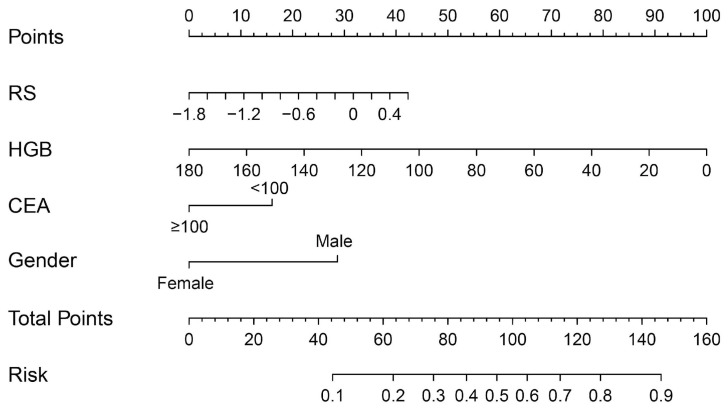
Radiomics nomogram to discriminate the primary tumours in patients with liver metastases.

**Figure 3 diagnostics-13-02937-f003:**
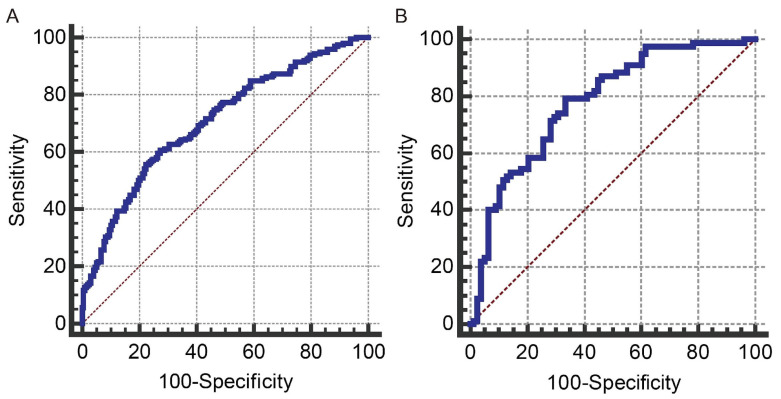
The received operating characteristics (ROC) curves for the prediction model in the training cohort (**A**) and the validation cohort (**B**). Blue line: ROC curve; brown line: reference line.

**Figure 4 diagnostics-13-02937-f004:**
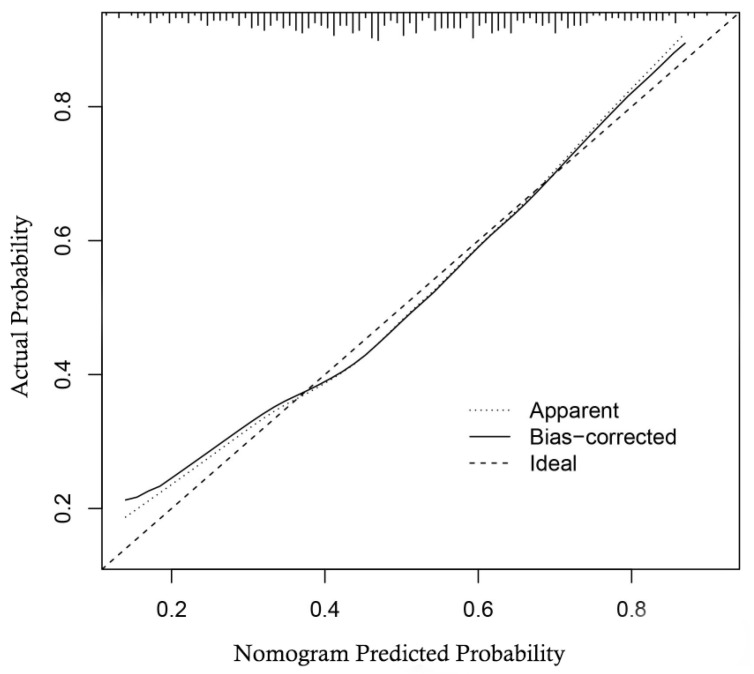
Calibration curve of the nomogram in the training cohort.

**Figure 5 diagnostics-13-02937-f005:**
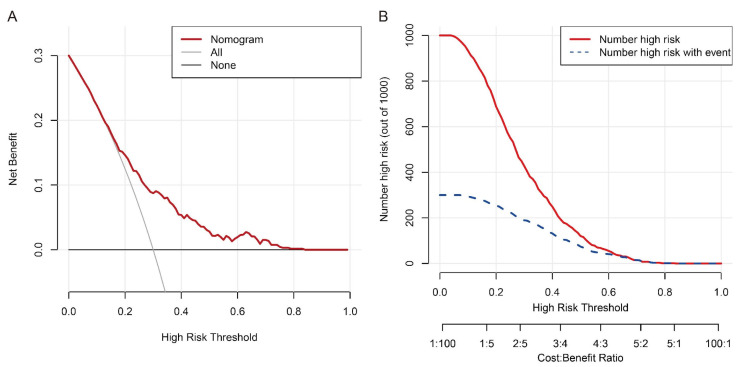
Decision curve analysis (**A**) and clinical impact curve (**B**) of the nomogram in the training cohort.

**Table 1 diagnostics-13-02937-t001:** Baseline characteristics of patients with liver metastases.

Characteristics	Training Cohort (*n* = 400)	Validation Cohort (*n* = 155)	*p*
Age (years)	67.0 ± 33.7	67.0 ± 9.6	0.914
Sex			0.691
Male	283 (70.8%)	107 (69.0%)	
Female	117 (29.3%)	48 (31.0%)	
Hypertension	126 (31.5%)	57 (36.8%)	0.236
Diabetes	51 (12.8%)	17 (11.0%)	0.566
Liver metastases			0.834
1	49 (12.3%)	20 (12.9%)	
>1	351 (87.8%)	135 (87.1%)	
Primary tumor			0.946
Gastric cancer	200 (50%)	77 (49.7%)	
Colorectal cancer	200 (50%)	78 (50.3%)	
HGB (g/L)	117 ± 23.9	116 ± 24.4	0.702
ALT (U/L)	18.1 ± 37.1	16.0 ± 30.5	0.368
Missing	3 (0.75%)	0	
CEA (ng/mL)			0.833
<100	287 (71.75%)	114 (73.5%)	
≥100	108 (27.0%)	41 (26.5%)	
Missing	5 (1.25%)	0	
CA19-9 (U/mL)			0.520
<1000	341 (85.3%)	137 (88.4%)	
≥1000	54 (13.5%)	28 (11.6%)	
Missing	5 (1.3%)	0	

HGB, blood haemoglobin; ALT, alanine transaminase; CEA, carcinoembryonic antigen; CA19-9, carbohydrate antigen 19-9.

**Table 2 diagnostics-13-02937-t002:** Baseline characteristics between patients with GCLMs and CCLMs.

Characteristics	Training Cohort			Validation Cohort		
	Gastric Cancer (*n* = 200)	Colorectal Cancer (*n* = 200)	*p*	Gastric Cancer (*n* = 77)	Colorectal Cancer (*n* = 78)	*p*
Age (years)	68.0 ± 9.1	65.5 ± 46.8	0.642	68.0 ± 9.0	67.5 ± 10.3	0.421
Gender			<0.001			<0.001
Male	160 (80%)	123 (61.5%)		66 (85.7%)	41 (52.6%)	
Female	40 (20.0%)	77 (38.5%)		11 (14.3%)	37 (47.4%)	
Hypertension	57 (28.5%)	69 (34.5%)	0.196	51 (66.2%)	47 (60.3%)	0.440
Diabetes	23 (11.5%)	28 (14.0%)	0.454	65 (84.4%)	73 (93.6%)	0.068
Liver metastases			0.286			0.354
1	21 (10.5%)	28 (14.0%)		8 (10.4%)	12 (15.4%)	
>1	179 (89.5%)	172 (86.0%)		69 (89.6%)	66 (84.6%)	
HGB (g/L)	112 ± 25.4	121 ± 21.1	<0.001	112 ± 24.7	122 ± 23.5	0.053
ALT (U/L)	18.2 ± 41.9	17.9 ± 31.6	0.149	17.6 ± 39.7	14.7 ± 15.3	0.021
CEA (ng/mL)			0.039			0.007
<100	153 (77.3%)	134 (68.0%)		64 (83.1%)	50 (64.1%)	
≥100	45 (22.7%)	63 (32.0%)		13 (16.9%)	28 (35.9%)	
CA19-9 (U/mL)			0.233			0.125
<1000	175 (88.4%)	166 (84.3%)		65 (84.4%)	72 (92.3%)	
≥1000	23 (11.6%)	31 (15.7%)		12 (15.6%)	6 (7.7%)	
Radiomics signature	−0.242 (0.349)	−0.316 (0.328)	0.010	−0.360 (0.303)	−0.322 (0.270)	0.039

GCLMs, liver metastases from gastric cancer; CCLMs, liver metastases from colorectal cancer; HGB, blood haemoglobin; ALT, alanine transaminase; CEA, carcinoembryonic antigen; CA19-9, carbohydrate antigen 19-9.

**Table 3 diagnostics-13-02937-t003:** Principal parameters calculated by texture analysis of liver metastases.

Texture Parameter	Colorectal Cancer (*n* = 200)		Gastric Cancer (*n* = 200)		*p*
	Mean	SD	Mean	SD	
Histogram					
Skewness	0.496	0.140	0.473	0.140	0.097
Kurtosis	0.211	0.155	0.232	0.128	0.398
Entropy	0.554	0.206	0.557	0.217	0.890
Energy	0.196	0.142	0.201	0.159	0.725
GLCM					
Homogeneity	0.495	0.144	0.521	0.147	0.078
Energy	0.088	0.119	0.083	0.125	0.642
Contrast	0.160	0.128	0.142	0.111	0.139
Correlation	0.672	0.175	0.668	0.190	0.852
Entropy	0.331	0.225	0.340	0.216	0.677
Dissimilarity	0.280	0.146	0.257	0.135	0.096
GLRLM					
SRE	0.550	0.161	0.536	0.173	0.382
LRE	0.414	0.163	0.429	0.176	0.376
LGRE	0.096	0.102	0.085	0.091	0.251
HGRE	0.418	0.193	0.429	0.201	0.582
SRLGE	0.119	0.125	0.106	0.115	0.274
SRHGE	0.418	0.192	0.427	0.198	0.636
LRLGE	0.047	0.053	0.041	0.042	0.186
LRHGE	0.395	0.174	0.410	0.189	0.413
GLNU	0.038	0.086	0.038	0.061	0.920
RLNU	0.058	0.103	0.057	0.084	0.957
RP	0.536	0.167	0.521	0.179	0.385
NGLDM					
Coarseness	0.204	0.167	0.195	0.153	0.556
Contrast	0.034	0.091	0.028	0.075	0.481
Busyness	0.0850	0.099	0.080	0.073	0.583
GLZLM					
SZE	0.640	0.115	0.629	0.128	0.360
LZE	0.308	0.146	0.321	0.160	0.399
LGZE	0.149	0.150	0.136	0.151	0.377
HGZE	0.428	0.194	0.438	0.201	0.638
SZLGE	0.138	0.143	0.129	0.151	0.517
SZHGE	0.407	0.180	0.410	0.179	0.849
LZLGE	0.021	0.028	0.018	0.018	0.159
LZHGE	0.325	0.162	0.348	0.188	0.198
GLNU	0.044	0.091	0.043	0.067	0.922
ZLNU	0.074	0.117	0.073	0.100	0.965
ZP	0.245	0.095	0.240	0.105	0.565

GLCM, the grey-level co-occurrence matrix; GLRLM, the grey-level run length matrix; NGLDM, the neighbourhood grey-level different matrix; GLZLM, the grey-level zone length matrix; SRE, Short-Run Emphasis; LRE, Long-Run Emphasis; LGRE, Low Gray-level Run Emphasis; HGRE, High Gray-level Run Emphasis; SRLGE, Short-Run Low Gray-level Emphasis; SRHGE, Short-Run High Gray-level Emphasis; LRLGE, Long-Run Low Gray-level Emphasis; LRHGE, Long-Run High Gray-level Emphasis; GLNU, Gray-Level Non-Uniformity for run; RLNU, Run Length Non-Uniformity; RP, Run Percentage; SZE, Short-Zone Emphasis; LZE, Long-Zone Emphasis; LGZE, Low Gray-level Zone Emphasis; HGZE, High Gray-level Zone Emphasis; SZLGE, Short-Zone Low Gray-level Emphasis; SZHGE, Short-Zone High Gray-level Emphasis; LZLGE, Long-Zone Low Gray-level Emphasis; LZHGE, Long-Zone High Gray-level Emphasis; GLNU, Gray-Level Non-Uniformity for zone; ZLNU, Zone Length Non-Uniformity; ZP, Zone Percentage.

**Table 4 diagnostics-13-02937-t004:** Multivariable analysis in the training cohort.

	Multivariable Analysis		
Variables	OR	95%CI	*p*
Sex			
Female	REF		
Male	3.457	2.102–5.684	<0.001
HGB	0.976	0.967–0.986	<0.001
CEA (ng/mL)			
<100	REF		
≥100	0.500	0.307–0.814	0.005
Radiomics signature	2.147	1.127–4.091	0.020

HGB, blood haemoglobin; CEA, carcinoembryonic antigen; REF, reference.

## Data Availability

The datasets utilised and/or analysed during the current study are available from the corresponding author upon reasonable request.

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
