# Peer review of "Development and Validation of a Radiomics Nomogram for Liver Metastases Originating from Gastric and Colorectal Cancer"

_diagnostics, 2023, doi:10.3390/diagnostics13182937_

Round 1
Reviewer 1 Report
Comments and Suggestions for Authors
In this study, the radiomics signature was well classified through nomogram for colorectal cancer and gastric cancer patients with liver metastasis. Although the study is interesting, the following variables must be considered.
1. Classification by staging of primary cancer in all patients did not include. In the case of late stage of cancer patients, the possibility of new lesions occurring at about two-week intervals is high. How to control for these variables should be discussed.
2. Differences by therapeutic modality in all patients did not reflect. If regimens subjected to all patients were not performed identically, this result may not be useful as a reference for future clinical treatment. This point needs to be discussed.
Comments on the Quality of English LanguageThere is no big problem with the quality of English.
Author Response
Dear Editor,
Thank you for the thorough review of our manuscript, previously titled "Development and Validation of a Radiomics Nomogram for Liver Metastases Originating from Gastric and Colorectal Cancer," for potential publication in Diagnostics. We extend our appreciation to both you and the reviewers for providing valuable feedback. In response to your suggestions, we have made revisions to the manuscript, which are indicated in red. Additionally, we have attached a detailed point-by-point response outlining how we addressed the reviewers' comments.
We appreciate your continued consideration and any further input you may have. Please feel free to reach out to us with any questions or recommendations.
Sincerely,
Haifeng Shi
Response to Reviewer 1 Comments:
Point 1: Classification by staging of primary cancer in all patients did not include. In the case of late stage of cancer patients, the possibility of new lesions occurring at about two-week intervals is high. How to control for these variables should be discussed.
Response: Thank you for your guidance. Surgery is not a viable treatment option for nearly all patients with liver metastases originating from gastric cancer (GCLMs). Conversely, some patients with liver metastases originating from colorectal cancer (CCLMs) do undergo surgery after their initial diagnosis. Consequently, obtaining precise staging information for the primary cancer, particularly in GCLM cases, is often unfeasible. In our study, all enrolled patients with liver metastases were diagnosed at an advanced cancer stage.
Point 2: Differences by therapeutic modality in all patients did not reflect. If regimens subjected to all patients were not performed identically, this result may not be useful as a reference for future clinical treatment. This point needs to be discussed.
Response: This study aimed to identify the primary site of liver metastases originating from gastric or colorectal cancer using radiomic features extracted from pre-treatment CT images. Since these radiomic features were obtained prior to any treatment, therapeutic regimens did not influence the results and conclusions presented in this paper. Our focus was solely on investigating the diagnostic value of radiomic features in distinguishing primary cancer sites, not on predicting clinical outcomes.
Response to Reviewer 2 Comments:
Point 1: In abstract section, in the 1st sentence word "Purpose" should be deleted. "Abstract: Purpose: To develop and validate a radiomics nomogram to determine the primary…"
Response: Acknowledged. We have removed the subheadings and restructured the "Abstract" accordingly.
Point 2: There is no conclusion in the paper. Provide conclusion section (about 200-250 words) and summarized your work in this part. Support conclusion with results.
Response: Understood. Please refer to the "Conclusion" section for the revised content.
Point 3: The quality of all figures (1-4) should be improved.
Response: The resolution of Figure 1-4 has been increased from 300dpi to 600dpi.
Point 4: The provided "Scheme 1" in line 211 should be reported as "Figure" and quality of this figure should improve. The written text should be readable in printed version or 100% zoom.
Response: Completed. Scheme 1 has been revised to Figure 1, and its quality has been enhanced.
Point 5: The introduction should provide more background information on liver metastases from GC and CRC, including their incidence, prognosis, and current treatment options.
Response: Completed. Additional information about liver metastases from gastric cancer (GC) and colorectal cancer (CRC) has been included. Please refer to the "Introduction" section.
Point 6: The methodology section should provide more details on the selection criteria for patients and the radiomics feature extraction process.
Response: Completed. Please find the added information in the "Study Design and Patients" and "CT Image Acquisition and Image Processing" sections.
Point 7: The results section should include more detailed information on the performance of the radiomics nomogram, including sensitivity, specificity, positive predictive value, and negative predictive value.
Response: Completed. In the training cohort, the nomogram exhibits a specificity of 77.66% and a sensitivity of 55.56%. Its positive predictive value is 71.43%, and the negative predictive value is 63.48%. Meanwhile, in the validation cohort, the nomogram demonstrates a specificity of 66.7%, a sensitivity of 79.2%, a positive predictive value of 70.12%, and a negative predictive value of 76.47%.
Point 8: The paper would benefit from a more concise and clear writing style, with shorter sentences and paragraphs and fewer technical terms and jargon.
Response: Finished. We have utilized shorter sentences and paragraphs and have made an effort to avoid the use of overly technical vocabulary.
Reviewer 2 Report
Comments and Suggestions for Authors
This paper entitled “Development and Validation of a Radiomics Nomogram for Liver Metastases Originating from Gastric and Colorectal Cancer” aimed to develop and validate a radiomics nomogram to determine the primary site of liver metastases originating from. There are some comments listed as follows:
1- In abstract section, in the 1st sentence word “Purpose” should be deleted.
“Abstract: Purpose: To develop and validate a radiomics nomogram to determine the primary…”
2- There is no conclusion in the paper. Provide conclusion section (about 200-250 words) and summarized your work in this part. Support conclusion with results.
3- The quality of all figures (1-4) should be improved.
4- The provided “Scheme 1” in line 211 should be reported as “Figure” and quality of this figure should improve. The written text should be readable in printed version or 100% zoom.
5- The introduction should provide more background information on liver metastases from GC and CRC, including their incidence, prognosis, and current treatment options.
6- The methodology section should provide more details on the selection criteria for patients and the radiomics feature extraction process.
7- The results section should include more detailed information on the performance of the radiomics nomogram, including sensitivity, specificity, positive predictive value, and negative predictive value.
8- The paper would benefit from a more concise and clear writing style, with shorter sentences and paragraphs and fewer technical terms and jargon.
Comments on the Quality of English Language
Minor editing of English language required
Author Response
Point 1: In abstract section, in the 1st sentence word "Purpose" should be deleted. "Abstract: Purpose: To develop and validate a radiomics nomogram to determine the primary…"
Response: Acknowledged. We have removed the subheadings and restructured the "Abstract" accordingly.
Point 2: There is no conclusion in the paper. Provide conclusion section (about 200-250 words) and summarized your work in this part. Support conclusion with results.
Response: Understood. Please refer to the "Conclusion" section for the revised content.
Point 3: The quality of all figures (1-4) should be improved.
Response: The resolution of Figure 1-4 has been increased from 300dpi to 600dpi.
Point 4: The provided "Scheme 1" in line 211 should be reported as "Figure" and quality of this figure should improve. The written text should be readable in printed version or 100% zoom.
Response: Completed. Scheme 1 has been revised to Figure 1, and its quality has been enhanced.
Point 5: The introduction should provide more background information on liver metastases from GC and CRC, including their incidence, prognosis, and current treatment options.
Response: Completed. Additional information about liver metastases from gastric cancer (GC) and colorectal cancer (CRC) has been included. Please refer to the "Introduction" section.
Point 6: The methodology section should provide more details on the selection criteria for patients and the radiomics feature extraction process.
Response: Completed. Please find the added information in the "Study Design and Patients" and "CT Image Acquisition and Image Processing" sections.
Point 7: The results section should include more detailed information on the performance of the radiomics nomogram, including sensitivity, specificity, positive predictive value, and negative predictive value.
Response: Completed. In the training cohort, the nomogram exhibits a specificity of 77.66% and a sensitivity of 55.56%. Its positive predictive value is 71.43%, and the negative predictive value is 63.48%. Meanwhile, in the validation cohort, the nomogram demonstrates a specificity of 66.7%, a sensitivity of 79.2%, a positive predictive value of 70.12%, and a negative predictive value of 76.47%.
Point 8: The paper would benefit from a more concise and clear writing style, with shorter sentences and paragraphs and fewer technical terms and jargon.
Response: Finished. We have utilized shorter sentences and paragraphs and have made an effort to avoid the use of overly technical vocabulary.
Round 2
Reviewer 1 Report
Comments and Suggestions for Authors
All concerns have been well addressed. No additional concern to raise.
Comments on the Quality of English LanguageMinor language polishing is required.
Reviewer 2 Report
Comments and Suggestions for Authors
The authors have addressed most of my concerns.
Comments on the Quality of English LanguageMinor editing of English language required